# Programmed Cell Death Pathways in Cholangiocarcinoma: Opportunities for Targeted Therapy

**DOI:** 10.3390/cancers15143638

**Published:** 2023-07-15

**Authors:** Manuel Scimeca, Valentina Rovella, Valeria Palumbo, Maria Paola Scioli, Rita Bonfiglio, Gerry Melino, Mauro Piacentini, Luigi Frati, Massimiliano Agostini, Eleonora Candi, Alessandro Mauriello

**Affiliations:** 1Department of Experimental Medicine, TOR, University of Rome Tor Vergata, 00133 Rome, Italy; manuel.scimeca@uniroma2.it (M.S.); valeria.palumbo.25@students.uniroma2.eu (V.P.); mariapaola.scioli@alumni.uniroma2.eu (M.P.S.); rita.bonfiglio@uniroma2.it (R.B.); melino@uniroma2.it (G.M.); m.agostini@med.uniroma2.it (M.A.); candi@uniroma2.it (E.C.); 2Department of Systems Medicine, University of Rome Tor Vergata, 00133 Rome, Italy; valentina.rovella@uniroma2.it; 3Department of Biology, University of Rome Tor Vergata, 00133 Rome, Italy; mauro.piacentini@uniroma2.it; 4Institute Pasteur Italy-Cenci Bolognetti Foundation, Via Regina Elena 291, 00161 Rome, Italy; luigi.frati@uniroma1.it; 5IRCCS Neuromed S.p.A., Via Atinense 18, 86077 Pozzilli, Italy

**Keywords:** cholangiocarcinoma, programmed cell death, apoptosis, ferroptosis, pyroptosis, necroptosis, targeted therapy

## Abstract

**Simple Summary:**

Cholangiocarcinoma is a highly aggressive cancer that originates from the bile ducts. Traditional treatments have limited effectiveness, necessitating the exploration of new approaches. Recent studies have highlighted the importance of programmed cell death mechanisms, including apoptosis, ferroptosis, pyroptosis, and necroptosis, in the development and progression of cholangiocarcinoma. Targeting these cell death pathways may increase the susceptibility of cholangiocarcinoma cells to chemotherapy and immunotherapy. However, further research is necessary to fully understand the intricacies of programmed cell death in cholangiocarcinoma and potentially identify effective therapeutic strategies.

**Abstract:**

Cholangiocarcinoma is a highly aggressive cancer arising from the bile ducts. The limited effectiveness of conventional therapies has prompted the search for new approaches to target this disease. Recent evidence suggests that distinct programmed cell death mechanisms, namely, apoptosis, ferroptosis, pyroptosis and necroptosis, play a critical role in the development and progression of cholangiocarcinoma. This review aims to summarize the current knowledge on the role of programmed cell death in cholangiocarcinoma and its potential implications for the development of novel therapies. Several studies have shown that the dysregulation of apoptotic signaling pathways contributes to cholangiocarcinoma tumorigenesis and resistance to treatment. Similarly, ferroptosis, pyroptosis and necroptosis, which are pro-inflammatory forms of cell death, have been implicated in promoting immune cell recruitment and activation, thus enhancing the antitumor immune response. Moreover, recent studies have suggested that targeting cell death pathways could sensitize cholangiocarcinoma cells to chemotherapy and immunotherapy. In conclusion, programmed cell death represents a relevant molecular mechanism of pathogenesis in cholangiocarcinoma, and further research is needed to fully elucidate the underlying details and possibly identify therapeutic strategies.

## 1. Introduction

Cholangiolocarcinoma (CCA) is a cancer originating from cholangiocytes that can occur in any part of the biliary tree. It is often challenging to diagnose based on histopathologic analysis alone. CCA has a slow onset and tends to be highly invasive, spreading rapidly to nearby organs and lymph nodes [1]. As a result of its initially asymptomatic nature, most cases are not detected until they have reached an advanced stage. Despite advances in treatment options, the prognosis for patients with CCA remains very poor [2]. Furthermore, given the raised prevalence of several CCA risk factors in the western word, the incidence of CCA has steadily increased over the last decades [3]. Currently its incidence accounts for ~15% of all primary hepatic malignancies (second only to hepatocellular carcinoma (HCC)) and ~3% of gastrointestinal tumors [4,5,6]. In this regard, diagnosis in the regions with the highest overall age-standardized incidence of CCA worldwide is often not confirmed (only 9% of liver cancer) by histological/cytological data. The 5-year survival is estimated at 7–20%, with a high recurrence rate after surgery [7,8,9,10]. Based on the anatomical location, three subtypes of CCAs are identified [11,12]: intrahepatic cholangiocarcinoma (iCCA), perihilar cholangiocarcinoma (pCCA) and distal cholangiocarcinoma (dCCA).

### Signaling Pathways and Therapeutical Approaches

All CCA subtypes are associated with different risk factors, biology, pathophysiology, clinical presentations, treatment and prognosis, as well as distinct epidemiological propensity [13,14,15,16]. Other factors associated with the risk of developing CCA are related to some risk factors of metabolic conditions, such as obesity, hypertension, DM2 [17,18,19], cigarette smoking and alcohol consumption [20]; the increase worldwide in these conditions over the last few decades could explain the increase in CCA rate [12,21]. Lastly, there is a large group of sporadic CCA not associated to any known risk factor, where genetic polymorphisms affecting the DNA repair system (MTHFR, TYMS, GSTO1, and XRCC1), detoxification enzymes (ABCC2, CYP1A2, and NAT2), drug resistance, inflammatory response and immunological surveillance (KLRK1, MICA, and PTGS2) could play a role [18].

Several somatic mutations have been identified in CCA affecting genes involved in cell fate and differentiation, proliferation, survival, and maintenance of genomic integrity [15]. According to this, emerging evidence suggests that the dysregulation of programmed cell death pathways, including apoptosis, ferroptosis, pyroptosis and necroptosis, plays a crucial role in CCA development and progression [22,23]. Several molecular mechanisms underlying the resistance to cell death have been identified in CCA, including the aberrant expression of apoptosis and necroptosis regulators, dysregulated ferroptosis, and altered metabolic pathways. Therefore, targeting these pathways has emerged as a promising therapeutic strategy for CCA. Several preclinical studies have shown that the pharmacological and genetic manipulation of cell death pathways can effectively induce tumor cell death and enhance chemosensitivity in CCA models.

Chemotherapy-induced cytotoxicity in CCA involves a complex interplay of molecular mechanisms that ultimately lead to cellular death [24,25]. One of the primary mechanisms is through the activation of apoptotic pathways. Chemotherapeutic agents, such as cisplatin and gemcitabine, can trigger a cascade of events that culminate in the activation of caspases, which are key executioners of apoptosis [26,27]. These caspases, particularly caspase 3, play a crucial role in cleaving various cellular substrates, including structural and functional proteins, leading to cellular dismantling and programmed cell death [28]. Additionally, chemotherapy can induce DNA damage, resulting in the activation of DNA repair pathways or, if the damage is irreparable, triggering apoptosis through p53-mediated signaling [29,30]. Another mechanism involved in chemotherapy-induced cytotoxicity is the generation of reactive oxygen species (ROS) within cancer cells [29,30]. Increased ROS levels disrupt cellular redox balance and can cause oxidative stress, leading to DNA damage, lipid peroxidation, and mitochondrial dysfunction, ultimately promoting cell death. Understanding these cellular death mechanisms induced by chemotherapy in CCA is essential for optimizing treatment strategies and exploring novel therapeutic approaches.

The scientific rationale of this review is based on the growing body of evidence suggesting that the dysregulation of programmed cell death mechanisms, particularly apoptosis, ferroptosis, pyroptosis and necroptosis, plays a critical role in the pathogenesis and progression of CCA [19]. Thereby, a comprehensive review of the current state of knowledge regarding the molecular mechanisms underlying programmed cell death in CCA can provide valuable insights into the potential therapeutic opportunities and challenges in this field.

## 2. Apoptosis Regulation in Cholangiocarcinoma

Apoptosis refers to a genetically controlled cell death program that is extremely important for organisms in balancing the proliferation and elimination of old/damaged cells. Apoptosis was first described in 1972 in *C. elegans* [31]. Apoptotic cells are characterized by very specific morphological and molecular characteristics, including reductions in cell dimensions, nuclear and chromatin fragmentation, vacuolization, and the presence of apoptotic bodies [29,30,32]. The activation of caspases is an unequivocal marker of activated apoptosis. This mechanism is essential for homeostasis and can be activated in both physiological and pathological processes. In the first case, each living organism has a homeostatic condition that must be preserved and needs to balance cellular proliferation in tissues. Indeed, apoptosis is activated since embryogenesis and occurs any time cells lose their utility within the organism (e.g., inflammatory cells are killed after inflammation is resolved). Several pathological stimuli are also able to activate apoptotic processes, mainly in the presence of damaged cells. However, cancer cells frequently acquire the capability to evade the apoptosis program, thus undergoing uncontrolled growth [33]. Recent studies associated the occurrence of apoptosis with both the origin and progression of CCA [28,34,35]. Key concepts about apoptosis are listed in Box 1.

Box 1Key points of apoptosis.
**Key Points—References**
Apoptosis is a genetically controlled cell death program that regulates the balance between the proliferation and elimination of old/damaged cells—[29]Apoptosis is morphologically characterized by a reduction in cell dimensions, nuclear and chromatin fragmentation, vacuolization, and the formation of apoptotic bodies—[32]Several pathological stimuli activate apoptotic processes mainly in the presence of damaged cells, although cancer cells frequently undergo apoptosis—[33]In CCA cells an imbalance between pro-survival and pro-apoptotic factors often occurs, which leads to cancer cells’ survival and the activation of apoptosis in immune cells within the tumour microenvironment—[28,34]


### 2.1. Main Signaling Pathways

Table 1 summarizes the molecules involved in the main apoptotic signaling pathway in CCA.

Two major pathways lead to apoptotic cells: the extrinsic and intrinsic pathways. The activation of apoptosis through the extrinsic pathway requires the identification of molecular signals on the cytoplasmic membrane of damaged cells. These signals include molecules such as FasL or tumor necrosis factor (TNFα), which interact with their specific receptors, typically belonging to the TNF receptor family, containing a cytoplasmic domain called the “death domain” [36,37]. When FasL interacts with Fas receptors, the activated receptors form a trimer, which interacts with the protein FADD (Fas-associated death domains), recognizing the cytoplasmic death domains of receptors. In the case of TNF-α/TNFR1 interactions, after receptor oligomerization, the cytoplasmic death domains are bound by TRADD, which recruits FADD and RIP [38,39]. The assembly of this complex leads to the interaction and activation of pro-caspase 8, which in turn activates Cas3, by cleaving pro-caspase 3. When Cas3 is activated, the apoptotic effector pathway is triggered [40,41].

In CCA, cells express functional Fas and FasL reciprocally, with the activation of the NF-κB pathway, which induces a down-regulation and up-regulation in Fas and FasL expression, respectively [42]. This up-regulation of FasL and down-regulation of Fas may be responsible for tumor cell survival and the negative modulation of immune surveillance. Moreover, another study demonstrated that iCCA cells can modulate the activity of immune cells, mainly through the activation of the Fas/FasL pathway, which induces the apoptotic process in T and NK cells [43]. In this study, the up-regulation of c-FLIP, an inhibitor of extrinsic apoptosis activated by inflammatory cells, was found in human iCCA cells. CCA cells also respond to TNF-α apoptotic stimuli [44]. In this case, CCA cells can modulate inflammatory and immune responses by down-regulating TNF-α secretion through tumor-secreted exosomes [45].

Another apoptotic mechanism is represented by the intrinsic or mitochondrial-dependent pathway [46]. In some cases, a communication between intrinsic and extrinsic pathways occurs in the regulation of apoptosis. In response to extrinsic apoptotic signals, cells may require a caspase-8-mediated apoptotic amplification phase, where the BH3-only protein Bid is cleaved to generate the activated fragment t-Bid, which triggers the activation of the intrinsic pathway [46]. Normally, the intrinsic pathway is triggered by inner signals, such as oxidative stress, hypoxia, high Ca2+ concentrations, or genetic damage, leading to the activation of pro-apoptotic members and the inactivation of anti-apoptotic members of the Bcl-2 family [46]. These pro- and anti-apoptotic proteins interact with each other to maintain a balance within the cell. However, when the pro-apoptotic stimuli become excessive, the balance is disrupted, leading to the activation of pro-apoptotic proteins. The unbalanced pro-apoptotic BH3-only proteins such as Bak and bax can enter the outer mitochondrial membrane, causing MOMP and the release of mitochondrial proteins, such as cytochrome c, Smac/DIABLO, AIF, and endonuclease G. Once cytochrome c is released into the cytosol, it binds to Apaf-1, leading to the formation of an apoptosome complex. The CARD domain of Apaf-1 recruits pro-caspase-9, which undergoes auto-activation and subsequent proteolysis. Activated caspase-9 then activates executor caspases-3, -6, and -7, causing the cleavage of enzymatic substrates and apoptotic cell death [46,47].

Several studies have shown that anti-apoptotic Bcl-2 family proteins play a role in survival and resistance to chemotherapy in CCA. Indeed, when CCA cells become malignant, they frequently express anti-apoptotic factors such as Bcl-XL and Mcl-1, which can activate survival responses [48].

The expression of Bcl-XL and Mcl-1 in CCA cells inhibits apoptosis, contributing to the low efficacy of chemotherapy and radiotherapy, and promoting cancer cell survival. However, the expression of these molecules also exposes cancer cells to novel therapeutic targets that could complement traditional therapies and promote a coordinated effect in activating cell death. According to preclinical studies conducted in vitro, inhibiting Bcl-xL and Mcl-1 could enhance the effectiveness of standard chemotherapy and lead to synergistic cell death. Consequently, Bcl-xL has been identified as a potential target for the treatment of CCA [49]. The same study also revealed that a high expression of Bcl-xL is a positive prognostic factor in CCA, particularly for the perihilar subtype, as it indicates better survival rates. Another investigation evaluated the significance of mortalin, Bcl-2, and Bax as prognostic factors in iCCA, and found that an increased level of Bcl-2 and mortalin and the down-regulation of Bax were linked to anti-apoptotic effects and tumor progression in iCCA [50]. Furthermore, the study demonstrated that patients with high levels of expression of both mortalin and Bcl-2 and a low expression of Bak had significantly lower 2- and 5-year OS rates and cumulative recurrence rates compared to patients with different combinations of expression levels. The work of Yamashita et al. also supports the increased expression of anti-apoptotic factors in CCA [51]. In this study, the enrolment of patients with iCCA who were subjected to hepatic resection allowed them to assess the expression levels of somatostatin receptor 2 (SSTR2) and Bcl2 proteins to differentiate between perihilar (large bile duct) and peripheral (small bile duct) carcinogenesis in iCCA patients based on the expression levels of these two proteins. In this case, Bcl2 expression was found to be present only in small bile ducts and correlated with cancer tissues.

Additionally, apoptosis can be activated by a third pathway that involves immune cells directly. In fact, the immune-mediated cellular response plays a pivotal role in maintaining homeostasis, eliminating pathogens, and preventing the development of malignancies [52]. In recent years, the field of immunotherapy has gained significant attention as a promising approach for treating various diseases, including cancer. One crucial aspect of the immune-mediated cellular response is apoptosis. In the context of immunotherapy, programmed cell death pathways have garnered significant interest due to their potential as targets for therapeutic intervention [53,54,55]. The discovery of immune checkpoint inhibitors, such as antibodies against programmed cell death protein 1 (PD-1) and its ligand PD-L1, has revolutionized cancer treatment by unleashing the immune system’s ability to recognize and destroy tumor cells [56,57]. By blocking the inhibitory signals mediated by PD-1/PD-L1 interactions, these therapies restore the effector functions of cytotoxic T cells and enhance the immune-mediated cellular response against cancer cells. When the adaptive immune response is triggered, CD8-positive T cells and NK cells can induce their cytotoxic effects on infected or cancer cells via the granzyme/perforin apoptotic pathway [58]. This process requires the secretion of perforins, transmembrane pore-forming molecules that create pores in target cells, leading to the release of cytoplasmic granules containing the granzyme molecules. Once granzyme reaches the target cells, it promotes apoptotic activation. Granzyme B can directly, or through caspase-10, promote the proteolytic activation of pro-caspase 3, and subsequently the apoptotic execution pathway. On the other hand, when Granzyme A enters targeted cells, it reaches the mitochondria, where it triggers the cleavage of NDUFS3 in electron transport complex I, disrupting mitochondrial metabolism and generating ROS. These oxidative stress molecules promote the mobilization of the endoplasmic reticulum-associated SET complex into the nucleus, activating single-stranded DNA damage.

However, in CCA, the tumor microenvironment enhances immune checkpoint modulation, which negatively modulates the immune system and reduces the activity of cytotoxic cells, leading to the proliferation of the tumor and its escape from cell death through the inhibition of NK and cytotoxic T cells [59]. Figure 1 shows the main molecular pathways involved in the apoptosis of CCA.

**Table 1 cancers-15-03638-t001:** Main cell death molecules involved in CCA.

Molecules	Biological Function	Type of Cell Death	Pathway	References
Fas	TNF receptor family of proteins, containing a cytoplasmatic protein-binding regions called Death Domain (DD). After its binding with FasL, trimerization of activated receptors and FADD recruitment occurs.	Apoptosis	Extrinsic	[36]
FasL	Transmembrane protein of the TNF family. Induce apoptosis following the interaction with its receptor Fas.	Apoptosis	Extrinsic	[36]
FADD	Adaptor molecule that interacts with various cell surface receptors such as Fas or TNFR1 and mediates cell apoptotic signaling. It recruits and activates pro-caspase8.	Apoptosis	Extrinsic	[36]
TNFα	Extracellular cytokine able to activate inflammation, proliferation, and apoptosis. It interacts with TNFR1 to activate apoptotic signaling.	Apoptosis	Extrinsic	[37]
TNFR1	Cell membrane receptor, which binds soluble TNF. Its intracellular region contains a DD, involved in homo- and hetero-typic interactions with other DD-containing proteins.	Apoptosis	Extrinsic	[37]
TRADD	Adaptor molecule that interacts with DDs of Fas or TNFR1, forming a complex with RIP.	Apoptosis	Extrinsic	[37]
RIP	This protein interacts with TRADD or FADD on cytoplasmatic DDs of activated receptors to form a complex that activates caspase-activating cleavage.	Apoptosis	Extrinsic	[37]
Caspase-8	Pro-caspase8 is recruited by FADD and activated by lytic cleavage. Fas, FADD and caspase-8 interact and form the so called DISC (death-inducing signaling complex).	Apoptosis	Extrinsic	[37]
Caspase-3	Effector of apoptotic pathway, activated by cleavage after Cas-8 activation.	Apoptosis	Extrinsic and intrinsic	[37]
Bid	Activated by caspase-8 cleavage, with the release of c-Bid or t-Bid. It interacts with Bax, promoting the MOMP and apoptosis.	Apoptosis	Intrinsic	[44]
Bax	Pro-apoptotic mediator. It promotes cytochrome c release from mitochondria through permeabilization of the outer mitochondrial membrane.	Apoptosis	Intrinsic	[44]
Bak	Pro-apoptotic mediator. It promotes cytochrome c release from mitochondria through permeabilization of the outer mitochondrial membrane.	Apoptosis	Intrinsic	[44]
Bcl-XL	Anti-apoptotic mediator. It inhibits the activator Bid or other BH3-only proteins and Bax/Bak, by mutual sequestration.	Apoptosis	Intrinsic	[44]
Mcl-1	Anti-apoptotic mediator. It inhibits BH3-only proteins and Bax/Bak pore formation.	Apoptosis	Intrinsic	[44]
Bcl-2	Anti-apoptotic mediator. It inhibits BH3-only proteins and Bax/Bak pore formation.	Apoptosis	Intrinsic	[44]
cytochrome c	Cytochrome c is released by mitochondria as an apoptotic signal. It interacts with Apaf-1 and pro-caspase 9, resulting in the formation of the apoptosome complex.	Apoptosis	Intrinsic	[32]
Smac/DIABLO	These proteins are released from the intermembrane space of mitochondria into the cytosol. Their interactors are multiple IAPs (inhibitor apoptosis proteins), which are removed to activate both initiator and effector caspases.	Apoptosis	Intrinsic	[45]
Apaf-1	This protein contains caspase recruitment domain (CARD) in its N-terminal. It interacts with cytochrome c and recruits pro-caspase 9, forming the apoptosome complex and promoting activated caspase-9 formation by cleavage.	Apoptosis	Intrinsic	[32]
Caspase-9	Activated caspase-9 promotes the cleavage and activation of apoptotic caspase effector, caspase-3.	Apoptosis	Intrinsic	[32]
Caspase-6	Apoptotic caspase effector.	Apoptosis	Intrinsic	[32]
Caspase-7	Apoptotic caspase effector.	Apoptosis	Intrinsic	[32]
Granzyme B	Serine protease characterized by a perforin-dependent pro-apoptotic function triggered in infected or cancer cells by cytotoxic immune cells. Activates apoptosis by interacting directly with caspases or by cleaving Bid.	Apoptosis	Immune cells activated apoptosis	[58]
Granzyme A	Granzyme A cleaves proteins at sites after basic amino acids.	Apoptosis	Immune cells activated apoptosis	[58]
Perforin	Granule protein released by cytotoxic cells, which forms membrane pores on targeted cells.	Apoptosis	Immune cells activated apoptosis	[58]

### 2.2. Potential Therapeutical Approaches

There are various approaches available for the treatment of cholangiocarcinoma (CCA), with some therapies involving the activation of apoptosis, such as chemotherapy. According to a recent study of Shanshan et al. [60], a combined chemotherapeutic regimen of gemcitabine and anlotinib promotes apoptosis in vitro in intrahepatic CCA (iCCA) cell lines. Additionally, targeted therapies are available, including those targeting Isocitrate Dehydrogenase (IDH) 1 and 2 (frequently occurring in iCCA), FGFR, the MAPK pathway, HER family receptors, BRCA, and BRCA proteins. Another class of drugs employed in CCA treatment is immunotherapy, which act mainly by inhibiting immune checkpoints such as PD-1 or CTLA4. Nowadays, there is a growing need to identify good early diagnostic and/or prognostic markers for every cholangiopathy in which CCA is present, as well as potential targets for pharmacological therapy. MicroRNAs have emerged as promising candidates in this regard [61].

Numerous studies highlight the implication of microRNAs (miRNAs) in cancer research, as they play an important role in regulating cancer cells and impacting various aspects of cancer progression, including apoptosis induction, cell cycle regulation, metastasis promotion, and angiogenesis stimulation [30,62,63,64,65,66,67,68,69,70,71,72,73,74,75,76,77,78]. In a recent investigation by Xiong et al. [79], the effects of circSETD3, a circular RNA containing multiple miRNA binding sites, were studied in CCA cells. The experimental research also focused on miR-421, which was described to promote cell proliferation in human gastric cancer. Interestingly, although miR-421 has been revealed, by targeting MTA1, to suppress the proliferation and metastasis of colorectal cancer, it also has a role as an oncogenic miRNA in biliary tract cancer [80,81]. Specifically, the authors investigated a new therapeutic target for CCA, namely, circSETD3, which was found to regulate the miR-421/BMF axis and inhibit proliferation, while inducing apoptosis [80,81]. They analyzed the expression of miR-421 and circSEDT3 in CCA tissues and cell lines (HUCCT1, TFK1, and QBC939), using qRT-PCR analysis, finding that circSETD3 was down-regulated while miR-421 was up-regulated in CCA cells and tumor tissues. Furthermore, they transfected a circSETD3 plasmid into the TFK1 cell line, which significantly promoted the expression of circSETD3 and suppressed miR-421 expression. Using flow cytometry analysis, detection of caspase-3 activity, and an MTT assay for cell viability evaluation, the authors found that the circSETD3 plasmid inhibited cell proliferation and induced more apoptotic cells. They also observed an increase in caspase-3 activity, BAX, and cleaved Caspase3 expression, and a decrease in BCL2 levels in TFK1 cells compared with the control plasmid. Therefore, circSETD3 may be a promising therapeutic target for CCA treatment.

In a study focused on the role of miR-373, it was demonstrated that its overexpression promoted apoptosis in CCA cells by targeting ULK1 (unc-51 like autophagy activating kinase 1) [82]. The study utilized several CCA cell lines including RBE, QBC939, HCCC9810, HUCCT-1, and HIBEpiC. The overexpression of miR-373 and ULK1 significantly improved the expression of Bcl-2 and decreased the expression of Bax, Caspase-3, and Caspase-9, compared to only miR-373 overexpression [83]. Table 2 reports the main microRNAs suggested as promising therapeutic targets for CCA treatment.

**Table 2 cancers-15-03638-t002:** Promising microRNAs as therapeutic targets for CCA.

Molecule	Function	References
circSETD3	It is a circular RNA containing multiple miRNA binding sites. circSETD3 has been implicated in CCA progression.	[79]
miR-421	It promotes cell proliferation in human gastric cancer and represents a promising therapeutic target for CCA treatment.	[80,81]
miR-373	Its overexpression promotes apoptosis in CCA cells by targeting ULK1.	[82]
miR-191	It is involved in the initiation and progression of CCA.	[83]

In conclusion, the mentioned studies demonstrate the potential use of miRNAs as biomarkers for the diagnosis and prognosis of CCA, as well as their potential as therapeutic targets. Further research and analysis of miRNAs could lead to an improved understanding of the CCA’s pathogenesis, and offer novel insights into the development of targeted therapies for CCA patients. Lastly, the identification and validation of miRNA biomarkers and therapeutic targets may increase the clinical management of CCA and ultimately improve patient outcomes.

## 3. Ferroptosis in Cholangiocarcinoma

The lack of activity of the GPx4 (glutathione peroxidase 4) enzyme, which catalyzes the GSH (glutathione)-dependent conversion of membrane hydroperoxides to alcohols, is associated with ferroptosis [84,85,86]. Ferroptosis occurs when two crucial conditions occur along with the inactivation of GPx4: (i) aerobic metabolism, which causes phospholipids to serve as the starting point for continuous hydroperoxide synthesis; (ii) the cellular iron pool’s potential to provide reduced iron [87]. Morphologically, the process of ferroptosis starts with cell contractions, which are defined by the growth of perinuclear lipids and their subsequent diffusion into the cytoplasm [29]. Ferroptosis can now be distinguished by several characteristics, such as morphological features (plasma membrane rupture, cell swelling, outer mitochondrial layer rupture, smaller mitochondria); biochemical traits (greater lipid peroxidation, iron accumulation, and decreased endogenous antioxidant activity of molecules); protein anomalies (up-regulation of acyl-CoA and ACSL4 synthetase long-chain family member 4; ChaC glutathione-specific gamma-glutamylcyclotransferase 1, CHAC1; transferrin receptor, TFRC; prostaglandin-endoperoxide synthase 2, PTGS2; ferritin degradation, ARNTL, GPx4, and voltage-dependent anion channel 2/3, VDAC2/3; aryl hydrocarbon receptor nuclear translocator-like); DAMPs release (damage-associated molecular patterns) (such as KRASG12D and HMGB1 (high-mobility group box-1), a mutated KRAS protein); and, lastly, genetic markers (NFE2L2 activation (Nuclear factor erythroid 2-like 2) and the overexpression of CHAC1 and PTGS2) [88]. Iron depletion can have a direct action against peroxidation, or more interestingly, in the case of tumors, can completely block lipid peroxidation by decreasing blood flow with the subsequent reduction in oxygen availability. Numerous studies are gradually revealing small compounds that might trigger or regulate ferroptosis. In this context, a recent study has demonstrated that the protective enzyme GPx4 inactivates the small molecule RSL3 through the adapter molecule 14-3-3ε [89]. A small molecule known as erastin has been found to specifically kill cells that overexpress the oncogenic proteins RAS, suggesting that it may be a strong inducer of ferroptosis [90].

In recent years, numerous approaches have been explored to stratify malignant CCA according to its prognosis, and identify potential targets for diagnosis and therapy [91,92]. It is discovered that the lipid peroxidase pathways play a role in CCA occurrence. Thus, the modulation of ferroptosis may represent an interesting avenue for the treatment of CCA. Obstructive jaundice caused by CCA determines the direct interaction of bile fluid with tumor cells. In this scenario, the dysregulation of bile iron metabolism is frequently observed in CCA [93]. Pathophysiological conditions that affect bile acid balance result in defects in the antioxidant defense system, thereby increasing oxidative stress [94]. Phospholipid GPx (glutathione peroxidase), an upstream regulator of the ferroptosis process, and the depletion of cysteine, a substance often present in metal-binding sites, may serve as precursors to ferroptosis [95]. The increases in lipid-derived ROS and subsequent ferroptosis death are linked to modifications in GPx and tumor formation [96]. In CCA, tissues are enriched with several transcription factors linked to oncogenic pathways, including P53 and myc, whereas VEGFR and ROS are involved in the stimulation of TAF [97,98,99,100]. The early detection of CCA can be aided by examining bile and its components, which can act as indicators. Photodynamic therapy (PDT) is a newly recognized palliative treatment for unresectable extrahepatic CCA that uses ROS to target tumors without causing cross-resistance to chemotherapy [101]. Chlorin A, a hydroporphyrin photosensitizer primarily found at the mitochondrial level, is used in PDT. Recent research suggests that Chlorin e6 and its derivatives function as CCA inhibitors [102,103]. Chlorin A accelerates ferroptosis in human CCA cell lines and is comparatively effective in comparison to temoporfin, a widely used photosensitizer to treat several tumor types. The PI3K/AKT/mTOR pathway, autophagosomal protein Beclin 1, and the phagosome-membrane elongating protein LC3-II levels are all increased by Chlorin A-PDT. Additionally, Chlorin A-PDT causes CCA cells derived from the autophagic pathway to undergo ferroptosis [104,105]. Ferroptosis-related genes serve as markers for survival, PDT sensitivity, and liver diseases [106,107]. Inhibition or activation of these pathways may significantly impact a CCA patient’s overall survival. Key concepts related to ferroptosis are listed in Box 2.

Box 2Key points of ferroptosis.
**Key Points—References**
Ferroptosis is a form of cell death characterized by the accumulation of reactive oxygen species (ROS) that result from iron-dependent lipid peroxidation—[87]During ferroptosis, the mitochondria undergo morphological changes, such as shrinkage, increased membrane density, and decreased mitochondrial crests. These changes are thought to be a result of lipid peroxidation and damage to the mitochondrial membranes, leading to the loss of membrane potential and mitochondrial dysfunction—[29]Rsl3 and Rsl5 are small molecules that have been identified as potent inducers of ferroptosis. They act by inhibiting the activity of glutathione peroxidase 4 (GPX4), a key regulator of lipid peroxidation, and lead to the increased accumulation of lipid reactive oxygen species (ROS) and subsequent cell death. The inhibition of GPX4 activity by Rsl3 or Rsl5 can also lead to a decrease in glutathione levels, which further enhances lipid peroxidation and ferroptosis. Therefore, these mediators are considered as negative regulators of GPX4 and promote iron-dependent oxidative cell death—[89]Slc7a11 has been shown to increase the resistance of cancer cells to ferroptosis, leading to tumor progression and therapy resistance in CCA—[89]Ferroptosis mediators are novel reliable targets for CCAs therapy—[106,107]


### 3.1. Main Signaling Pathways

Table 3 summarizes the molecules involved in the main signaling pathway of ferroptosis in CCA.

A potential target for CCA therapy is the phenomenon of ferroptosis. There are now small compounds or nanomaterials that can induce ferroptosis, which not only help overcome treatment resistance, but also prevent tumor spread. However, the functions of DEFRGs (differentially expressed ferroptosis-related genes) in CCA have not been well investigated and need further research [91,92,93,94,95,96].

Recently, Wang et al. [108] discovered the primary relationships between DEFRGs such as Rsl3 and rsl5 and tumor metabolism, oxidative stress, and ferroptosis in CCA. To assist clinicians in developing better treatment strategies, a nomogram based on risk scores and clinicopathological values was created [108]. The ROC curve showed that the model may be used for the earlier detection of CCA, and has excellent sensitivity and specificity. Additionally, single-sample immune cell enrichment analyses revealed a close association between ferroptosis and immunity in CCA. Thus, understanding the precise pathophysiology of CCA is crucial for the development of customized treatment. The DEFRGs were considerably enriched in pathways linked to iron ion reaction, ferroptosis, carbon metabolism, cell oxidative stress, ROS (reactive oxygen species), and other associated processes, according to GO and KEGG analyses.

Mitochondrial autophagy receptor BNIP3, a crucial protein in cells, is required to prevent the production of ROS [109,110,111,112]. BNIP3 is controlled by the upstream regulator HIF-1 (hypoxia-inducible factor), which is extensively expressed in healthy tissues and can prevent the development of many different types of tumors [113,114]. However, the relationship between inflammation and tumors promotes the formation of ROS in malignant cells, potentially leading to genetic instability, and oxidative and DNA damage [115,116,117,118,119]. The BNIP3 gene is now considered a risk factor for cancer and may be influenced by various molecules. The ALOX5 gene encodes a family of lipases that are essential to produce leukotrienes and involved in several activities, including inflammation and tumors [113]. Researchers studying stroke have suggested that N-acetylcysteine (NAC) may prevent ferroptosis by neutralizing the harmful lipids produced by ALOX5 [96]. However, further research is required to understand the function of ALOX5 in CCA. The primary regulator of microtubule dynamics, STMN1, is essential for controlling the cell cycle, which is strongly associated with the division and proliferation of tumor cells [120,121].

The association between immunity and tumor development has become clearer in recent years, and this is an area of active research. Several studies have demonstrated the importance of the tumor immune microenvironment in determining tumor prognosis and assessing treatment efficacy [122]. Different types of immune cell infiltration may provide an environment that is either favorable or unfavorable for the growth and spread of tumor cells [123]. Interestingly, some studies have shown that BNIP3 expression is upregulated in activated NK cells and is involved in promoting NK cell-mediated cytotoxicity against tumor cells [124]. BNIP3 expression has been associated with the increased production of cytotoxic molecules, such as perforin and granzymes, by NK cells. This suggests that BNIP3 may contribute to the anti-tumor immune response mediated by NK cells [125,126]. Notably, NK CD56-positive cells in head and neck cancer release IFN-γ to promote anticancer adaptive immunity, which supports this finding [127]. Also, it is well-recognized that the presence of neutrophils in tumors is frequently associated with a poor prognosis for patients because they promote growth, invasion, and immunosuppression of various tumors, and induce blood vessels by releasing RNS (reactive nitrogen species), ROS, or proteases to promote tumorigenesis [128,129,130]. This is consistent with the poor prognosis of patients with elevated BNIP3 expression in CCA.

Furthermore, a significant correlation between Th2 cells and the oncogene STMN1 in CCA has been described. High tumor malignancy is associated with a low Th1/Th2 ratio, as tumor cells release immunosuppressive prostaglandins and cytokines that favor Th2 cell activity and reduce antitumor potential [131,132]. Several immunological markers of Th2 cells include CDC25C, BIRC5, HELLS, CENPF, and NEIL3. BIRC5 is known to be crucial for cell division and may prevent apoptosis, while the overexpression of BIRC5 encourages tumor growth in hepatocellular carcinoma [133]. CDC25C significantly regulates the cell cycle and is highly expressed in prostate cancer [134]. The increased expression of NEIL3 and HELLS is associated with decreased survival time in liver and other cancers [135,136]. These results may explain why patients with elevated STMN1 expression in CCA have a poor prognosis.

Yao et al. [137] utilized RNA-seq to examine the number of immune cells and their impact on the prognosis of CCA, and to assess the function of different immune cell infiltrates in CCA. The scRNA-seq dataset (GEO: GSE138709) was used to identify the variety and heterogeneity of several cell subsets in CCA tissues. Immune cell infiltration was examined using CIBERSORT, and it was discovered that monocyte infiltration was associated with a better prognosis. An examination of the intercellular interaction between monocytes and cholangiocytes in CCA was conducted to determine the impact of monocyte infiltration on CCA, and it was shown that there was a strong correlation between TFRC and TNFSF13B. TFRC is a ferroptosis driver and marker that regulates ferroptosis by serving as a channel for Fe^3+^ in the outer membrane, while TNFSF13B is a monocyte cytokine indicator. Malignant epithelial cells were markedly enriched in ferroptosis signaling processes in comparison to benign epithelial cells, and TFRC was substantially overexpressed in the cluster 4 fractions. The authors propose that the TNFSF13B produced by monocytes induces ferroptosis by interacting with the TFRC receptor on cluster 4 and therefore facilitating the transfer of Fe^3+^. Using decision curve analysis (DCA) and receiver operating characteristic (ROC) curves, the accuracy of the signature in predicting CCA prognosis was assessed. To facilitate clinical application, the authors also created a nomogram risk evaluation map that included clinical indicators and risk scores. Overall, the study investigated how immune cell infiltration influenced the CCA prognosis, and used single cell sequencing to discover a potential mechanism by which monocytes may regulate malignant intracellular ferroptosis. The authors suggest that BNIP3, CENPW, and TMEM107 may be targeted for CCA.

A recent article by Lei and colleagues [138] elucidated the role of JUND/linc00976 in CCA development by inhibiting ferroptosis. By preserving the reduction of cystine to cysteine, linc00976 is known to potentially maintain GSH production. The TrxR/Trx system, which includes selenocysteine-dependent antioxidant enzymes, may play a role in some of this linc00976-dependent cysteine production [139]. Interestingly, linc00976 depletion significantly decreased folate biosynthesis [140]. This suggests that linc00976 may regulate this metabolic pathway at the transcriptional level. Additionally, it has been shown that the availability of tetrahydrobiopterin (BH4) regulates cellular responses to ferroptosis following GPX4 inhibition. Therefore, it is plausible that linc00976 may protect CCA cells against ferroptosis development independently of GPX4 by controlling folate synthesis at the transcriptional level. These findings suggest that targeting linc00976 could offer a potential treatment approach for CCA patients.

Yao and colleagues [141] utilized signature analysis to identify new ferroptosis-related genes in CCA by examining the accessible database. Among these, TP53 has been identified as a prognostic factor and potential therapeutic target. It has a dual role in ferroptosis in various cancers through multiple pathways [142,143,144,145]. Moreover, TP53, which is more expressed in CCA, acts as a protective factor. Regarding ACSL4, it was shown to be a critical factor in ferroptosis promotion [146]. In fact, increased ACSL4 expression was linked to a poorer prognosis for CCA. Inhibiting IREB2 significantly reduces ferroptosis in non-small-cell lung cancer, a common ferroptosis-related gene, which reduces anticancer efficacy [147]. The research by Yao and co-workers [141] also indicates the protective role of IREB2 for ICC, with higher IREB2 expression in tumor tissue linked to better overall survival. NFE2L2 is a crucial regulator of ferroptosis and a potential therapeutic target for neurodegenerative disorders. NFE2L2 was identified by Yi and colleagues [148] as a low-risk gene in many diseases, including bladder cancer. High NFE2L2 expression was also linked to improved prognosis in CCA.

Finally, several CCA SNPs related to ferroptosis have been identified. Ex vivo genetic analysis of 122 human liver biopsy samples revealed a specific negative correlation between IDH1105GGT levels and GPX4 expression, as well as SNPs in CCA [149]. This IDH1105GGT SNP may alter the stability or level of IDH1 mRNA, resulting in changes in NADPH production [150,151,152]. Thus, the IDH1105GGT SNP may be linked to the development of metastasis and CCA through the modification of GPX4 expression. Figure 2 shows the main molecular pathways of ferroptosis in CCA.

**Table 3 cancers-15-03638-t003:** Main molecules involved in ferroptosis.

Molecules	Biological Function	Type of Cell Death	References
glutathione peroxidase 4	GPX4 is an antioxidant enzyme that plays a crucial role in repairing oxidative damage to lipids and is a key inhibitor of ferroptosis.	Ferroptosis	[87]
RSL3	RSL3 is a small molecule compound that inhibits the activity of GPX4, an antioxidant defense enzyme. This inhibition leads to the accumulation of lipid peroxides and the induction of ferroptosis, a form of regulated cell death.	Ferroptosis	[89]
RSL5	It is a transcription factor that increases the expression of iron metabolism inhibitors such as ferritin light chain (FTL) and ferritin heavy chain 1 (FTH1).	Ferroptosis	[89]
p53	The role of p53 in ferroptosis is paradoxical, as it can have both pro- and anti-ferroptotic effects. On one hand, p53 can induce ferroptosis by inhibiting Solute carrier family 7 member 11 (SLC7A11), a component of the glutamate-cystine antiporter, which reduces intracellular cysteine levels and impairs glutathione synthesis, a crucial antioxidant. Additionally, p53 can induce ferroptosis by upregulating spermidine/spermine N1-acetyltransferase 1 (SAT1) or glutaminase 2 (GLS2), which leads to an increase in lipid peroxidation and reactive oxygen species (ROS) production.	Ferroptosis, apoptosis	[141]
BNIP3	It is a mitochondrial autophagy receptor involved in the production of ROS.	Ferroptosis	[125,126]
nicotinamide adenine dinucleotide phosphate (NADPH) Oxidase 4 (NOX4)	(NADPH) Oxidase 4 (NOX4) is an enzyme complex consisting of multiple subunits that use nicotinamide adenine dinucleotide phosphate (NADPH) as a substrate to generate reactive oxygen species (ROS), including superoxide anions. The excessive production of ROS by NOX4 has been shown to promote ferroptosis, a type of cell death characterized by the accumulation of lipid peroxides.	Ferroptosis, apoptosis	[150,151,152]
ALOX5	ALOX5 gene encodes a family of lipases that are essential for the production of leukotrienes and involved in several activities, including ferroptosis, inflammation and tumors.	Ferroptosis	[113]
STMN1	The primary regulator of microtubule dynamics STMN1 is essential for controlling the cell cycle, which is strongly associated with the division and proliferation of tumor cells.	Ferroptosis	[120,121]
TNFSF13B	TNFSF13B facilitates the transfer of Fe^3+^ by interacting with the TFRC receptor on cluster 4.	Ferroptosis	[137]
BH4	BH4 regulates cellular responses to ferroptosis following GPX4 inhibition.	Ferroptosis	[140]

### 3.2. Possible Therapeutical Approaches

CCA cells have been found to be particularly susceptible to ferroptosis, suggesting that inducing this type of cell death may selectively kill cancer cells while sparing normal cells. In this context, researchers investigated the use of ferroptosis inducers, such as erastin and RSL3, in preclinical models of CCA. These inducers have been shown to inhibit tumor growth as well as sensitizing cancer cells to chemotherapy and immunotherapy. In addition, the combination of ferroptosis inducers with other targeted therapies, such as inhibitors of the MEK/ERK pathway, has shown promising results in preclinical studies.

Overall, the therapeutic potential of ferroptosis in CCA is still in its early stages, and further research is needed to optimize its use. The discovery of ferroptosis as a potential therapeutic target for this aggressive cancer provides hope for the development of more effective treatments in the future.

## 4. Pyroptosis in Cholangiocarcinoma

Pyroptosis is a lytic and inflammatory form of cell death, also known as secondary necrosis, among the programmed cell death pathways. First described in the 1990s as an apoptotic form of pathogen-infected macrophages, it has since been defined as a non-apoptotic form of cell death. Morphologically, cell swelling, osmotic lysis, and the release of proinflammatory factors characterize pyroptotic process [153]. Like apoptosis, it is also characterized by nuclear condensation and DNA fragmentation, although the nuclear membrane maintains its integrity [154,155,156,157,158,159]. While emerging evidence suggests that pyroptosis may modulate the tumor microenvironment (TME) in several cancers and may have a role in cancer modulation, its role in CCA has not yet been studied in detail and is currently under investigation [160,161,162]. Key concepts related to pyroptosis are listed in Box 3.

Box 3Pyroptosis.
**Key Points—References**
Pyroptosis is a form of lytic and inflammatory cell death, which leads to inflammasomes activation and pro-inflammatory cytokines efflux—[157]Pyroptosis is characterized by cell membrane pore formation, followed by inflammatory necrosis and tissue injury—[158,159]Pyroptosis may modulate the TME and activate immune cells against cancer cells—[161]The delivery of chemotherapeutic drug methotrexate within CCA cells increased pyroptosis and triggered antitumor activity and tumor disruption—[161]


### 4.1. Main Signaling Pathways

Table 4 summarizes the molecules involved in the main signaling pathway of pyroptosis in CCA.

The pyroptosis process involves a variety of signal molecules, including damage-associated molecular patterns (DAMPs) and pathogen-associated molecular patterns (PAMPs), recognized by immune cells through pattern recognition receptors (PRRs). NOD-like receptors (NLRPs), AIM2-like receptors (AIM2), and TRIM family members (Pyrin) play a central role in triggering pyroptosis after recognition of a pathogen or cell damage and inflammatory signals [163].

There are two main mechanisms for activating pyroptotic processes: canonical and non-canonical inflammasome pathways [164,165]. The first one is triggered when PAMPs or DAMPs interact mainly with NLRP3, but also with NLRP1, NLRC4, and AIM2, following the interaction of these receptors with ASC proteins and oligomerization. NLRP3-activated oligomers interact with and cleave pro-caspase 1 into caspase-1, which itself cleaves and activate pro-IL-1β and pro-IL-18 into their activated forms. Moreover, caspase-1 also cleaves and activates GSDMD by freeing its N-terminal domain, which can oligomerize and form membrane pores [158]. The non-canonical inflammasome activation pathway is a caspase-1 independent pathway in which GSDMD is activated by caspase-4/5/11 and is generally activated by LPS [159]. Unlike caspase-1, caspase-4/5/11 cannot directly promote pro-IL-1β and pro-IL-18 maturation; however, GSDMD pore formation and consecutive potassium efflux induce canonical NLRP3 inflammasome formation and the cleavage of IL-1β and IL-18. Alternative pathways of pyroptosis can occur through the activation of caspase 3, which mediates the switching from apoptosis to pyroptosis, or granzyme A or B, leading to activation of other gasdermins, such as GSDME [166].

**Table 4 cancers-15-03638-t004:** Main molecules involved in pyroptosis.

Molecules	Biological Function	Type of Cell Death	Pathway	References
ASC	It is a component of the inflammasome complex, promotes Cas-1 activation.	Pyroptosis	Canonical	[159,165]
Caspase-1	Caspase-1 induces pyroptosis through activating cleavage of GSDMD, IL-1β and IL-18.	Pyroptosis	Canonical	[159,165]
IL-1β	It is an effector of inflammatory response pyroptotic cell lysis.	Pyroptosis	Canonical	[159,165]
IL-18	It is an effector of inflammatory response pyroptotic cell lysis.	Pyroptosis	Canonical	[159,165]
NLRP3	NLRP3 triggers the formation of the inflammasome complex after interaction with DAMPS, PAMPs and LPS.	Pyroptosis	Canonical	[159,165]
GSDMD	It forms cell membrane pores after N-terminal lytic activation and induces pyroptotic inflammatory lysis.	Pyroptosis	Canonical and non-canonical	[159]
Caspase-4	Caspase-4induces pyroptosis through GSDMD cleavage.	Pyroptosis	Non-canonical	[159]
Caspase-5	Caspase-5 induces pyroptosis through GSDMD cleavage.	Pyroptosis	Non-canonical	[159]
Caspase-11	Caspase-11 induces pyroptosis through GSDMD cleavage.	Pyroptosis	Non-canonical	[150]
GSDME	It forms a cell membrane pore.	Pyroptosis	Inflammasome non-dependent	[166]
Caspase-3	Caspase-3 mediates the switching from apoptosis to pyroptosis and promotes GSDME cleavage and activation	Pyroptosis	Inflammasome non-dependent	[166]
granzyme A	It catalyzes proteolytic activation of GSDMB.	Pyroptosis	Inflammasome non-dependent	[166]
granzyme B	It catalyzes proteolytic activation of GSDME.	Pyroptosis	Inflammasome non-dependent	[166]

In terms of the role of pyroptosis in the pathobiology of liver ducts, it has been shown that mainly NLRP3 activation could be linked to some cholangiopathies. Maroni et al. [167] found that Nlrp3 is expressed in reactive cholangiocytes, and the overexpression of NLRP3 in these cells results in increased levels of IL-18 and impaired function of the epithelial barrier.

The role of NLRP3 pyroptosis–inflammasome activation in inflammatory bile duct disorders has been proposed by Tian et al. [168]. In this study, it has been reported that in primary biliary cholangitis (PBC), the NLRP3 inflammasome process plays a crucial role in guiding macrophage activation through in vitro and in vivo experiments. This process is triggered by Gal3, which can be considered an initiator of inflammatory signaling, switching on the NLRP3 inflammasome and activating IL-17 proinflammatory cascades. Figure 3 shows the main molecular pathways of pyroptosis in CCA.

### 4.2. Possible Therapeutical Approaches

Given its role in the inflammatory response and activation of the innate immune system, pyroptosis may play an important role in modulating the tumor microenvironment (TME), and could be leveraged to activate “cold” immune tumors [168]. Li et al. [169] found that the pyroptosis index, measured through single-sample Gene Set Enrichment Analysis (ssGSEA), could predict immunotherapy response and survival outcomes in several cancers, including CCA, likely due to the link between pyroptosis and the modulation of tumor-infiltrating immune cells. By amplifying inflammatory responses and recalling innate immune cells, pyroptosis could be triggered and enhanced in the TME to attract and activate inflammatory cells. Wang et al. [170] demonstrated that chemotherapy drugs induce pyroptosis in cancer cells in vitro, mediated by caspase-3 activation and the subsequent cleavage of GSDME. Gao et al. [171] conducted the most detailed study on the role of pyroptotic cell death in CCA. Using tumor cell-derived microparticles (TMP), methotrexate was delivered into CCA cells, inducing GSDME-mediated pyroptosis. After inflammasome and pyroptosis activation, macrophages from CCA patients were activated, and neutrophils were recruited to the tumor site, where both cells could exert antitumor activity and tumor disruption. Despite these findings, more research is needed to fully understand the role of pyroptosis in CCA etiology, prognosis, and potential therapeutic targets.

## 5. Necroptosis in Cholangiocarcinoma

The Nomenclature Committee on Cell Death published a consensus paper in 2023 [19] providing nomenclature, definitions, and an updated classification of all known forms of cell death, including necroptosis. It is a regulated form of cell death that shows morphological features of necrosis, such as increased cytoplasmic translucency, organelle swelling, lysosomal membrane permeabilization, increased cell volume, and intact nucleus, which distinguishes it from apoptosis [172,173].

TNF, Fas ligand, interferon, dsRNA, ATP depletion, as well as processes that take place outside of cells such as the production of reactive oxygen species, calcium overload, and ischemia/reperfusion injury (IRI), can cause necroptosis [174,175].

Recent studies have linked necroptosis to the prognosis of CCA. Multiple signaling pathways related to necroptosis have been identified and proposed as potential targets for both the diagnosis and treatment of CCA. Key concepts about necroptosis are listed in Box 4.

Box 4Necroptosis.
**Key Points—References**
Necroptosis is characterized by morphological features of necrosis, such as increased cytoplasmic translucency, organelle swelling, lysosomal membrane permeabilization, increased cell volume, and intact nucleus, unlike apoptosis—[172]The most studied necroptosis pathway is the TNF-α/TNFR1 axis—[172]Ripk3, ripk1, and mlkl are responsible for the crucial event of necrosome formation—[175]Promoting necroptosis or manipulating its pathways has emerged as a promising therapeutic strategy for cancer treatment, as it offers an alternative mechanism for eliminating cancer cells that are resistant to other forms of cell death—[175]


### 5.1. Main Signaling Pathways

Table 5 summarizes the molecules involved in the main signaling pathway of Necroptosis in CCA.

The TNF-α/TNFR1 axis is the necroptosis pathway that has received the most attention in research. Upon TNFR1 activation, the formation of a complex, named complex I, occurs, which is composed of TRADD (TNFR-associated factor 2), RIPK1 (receptor-interacting protein kinase 1), cIAPs 1 and 2 (cellular inhibitors of apoptosis proteins), and the linear ubiquitin chain assembly complex, and has been shown to trigger the nuclear factor-κB (NF-κB) signaling pathway.

According to this, different signaling complexes can determine cell death or cell survival:cIAPs, after the polyubiquitination of RIPK1, promote the activation of the NF-κB pathway, which stimulates cell survival by activating genes encoding molecules that have cytoprotective functions [176];If NF-κB or its regulators are blocked, RIPK1 undergoes deubiquitination, facilitating necroptosis. Additionally, several drugs can directly enhance the deubiquitination of RIPK1 [177].

The link between Fas-associated protein with the death domain (FADD), TRADD, FLICE-inhibitory protein (FLIP), and pro-caspase-8 induces the formation of complex IIa. The activation of pro-caspase-8 into caspase-8 by the IIa complex leads to RIPK1-independent necroptosis [175,178].

**Table 5 cancers-15-03638-t005:** Main molecules involved in necroptosis.

Molecules	Biological Functions	Type of Cell Death	References
IFN-γ	Molecule capable of inducing necroptosis	Necroptosis	[174]
TNFα	Molecule capable of inducing necroptosis	Necroptosis	[174]
Fas ligand	Molecule capable of inducing necroptosis	Necroptosis	[174]
LPS	Molecule capable of inducing necroptosis	Necroptosis	[174]
RIPK1	It is crucial for the formation of complex IIb (RIPK1-RIPK3-MLKL)	Necroptosis	[178]
RIPK3	RIPK3 is a serine/threonine–protein kinase that activates necroptosis	Necroptosis	[178]
MLKL	MLKL is a pseudokinase that is involved in TNF-induced necroptosis	Necroptosis	[175]
TNFR1	When it becomes active, it enables the recruitment of three specific necroptosis related proteins known as TRADD, RIP1, and TRAF2	Necroptosis	[174]
TRAF2	It can protect cells, inhibiting necroptotic cell death by TNF-induced NF-kB activation	Necroptosis	[174,175]
NF-kβ	If NF-κB is blocked, RIPK1 undergoes deubiquitination and is released, which leads to the formation of molecular complexes that facilitate necroptosis	Necroptosis	[177]
TRADD	It is a target protein for TNF-induced necroptosis in the absence of RIPK1	Necroptosis	[174,175]
FADD	It recruits the initiator caspase-8, forming death-inducing signaling complex (DISC)	Necroptosis	[174,175]
cIAP1	cIAP1 adds ubiquitin molecules to NF-κB, which triggers the activation of kinase NIK and the suppression of non-canonical NF-κB signaling	Necroptosis	[151]
cIAP2	It ubiquitinates NF-kB, inducing kinase (NIK) to suppress non-canonical NF-kB signaling and RIPK1 to promote cell survival	Necroptosis	[175]
TAK1	This is a serine/threonine kinase that modifies RIPK1 through phosphorylation, which controls its association with RIPK3 and facilitates necroptosis. It is a part of the TAK1 complex	Necroptosis	[175]

Another event that can occur is binding between RIPK1 and RIPK3, mediated by NF-κB, cIAPs, TAK1, FADD, FLIP, and pro-caspase-8. This interaction promotes the formation of complex IIb, which plays a critical role in RIPK1-dependent apoptosis [175,178].

If caspase-8 is suppressed or blocked, the interaction between RIPK1, RIPK3, and MLKL results in the formation of the necrosome, also known as complex IIc. This complex plays a crucial role in the activation of necroptosis. Moreover, the phosphorylation of MLKL by RIPK3 leads to the oligomerization and translocation of the cytosol to the plasma membrane [175,179].

The oligomers of MLKL can cause the formation of a pore-forming complex on the cell membrane that induces permeabilization, or inhibits the action of calcium and sodium channels. The result is an increase in intracellular osmotic pressure, which leads to the opening of membrane pores. This event is viewed as one of the defining features of necroptosis [87]. Consequently, DAMPs are released, which play a role in the activation of the inflammatory response. These DAMPs are recognized by Toll-like receptors (TLRs) and receptors for advanced glycation end products (RAGE), which activate innate immunity and increase cytokine release.

In the context of cancer, dendritic cells can be directed and activated in the tumor’s proximity, easing the recognition and elimination of tumor antigens. Once activated, dendritic cells mature and travel to lymph nodes, where they present tumor antigens to naïve CD8+ T cells, leading to the activation and differentiation of cytotoxic T cells. Moreover, the priming of these types of cells occurs, leading to the specific targeting and attacking of tumor cells, which can aid in the elimination of the cancer [180].

Necroptosis can serve as a mechanism to block tumor progression in cells where apoptosis is not successful. However, the activation of the inflammatory response, mediated by DAMPs, can result in cell proliferation, angiogenesis, and metastasis in tumor cells [180,181]. Regarding the contribution of necroptosis in CCA, it should be noted that CCAs can present different clinical characteristics, genotypes, and biological behaviors, which can depend on the location, growth pattern, and histological differentiation of the tumor [182,183].

In a study conducted by Saeed and colleagues [184], it was discovered that the type of hepatic tumor is influenced by the hepatic microenvironment. The researchers used mice and two different types of oncogenic delivery to investigate this phenomenon. They found that the same substances that promote cancer induce the formation of intrahepatic cholangiocarcinoma (iCCA) when provided to mature hepatocytes. Furthermore, the authors found that a microenvironment dominated by necroptosis promotes iCCA development, whereas an apoptotic environment favors the development of HCC. The necroptotic environment was obtained by the presence of DAMPs, derived from necroptotic hepatocytes. The outcome was the production of specific inflammatory cytokines secreted by immune cells [175,185,186]. On the contrary, hepatocytes that reside in a non-necroptotic microenvironment promote the formation of HCC or a combination of iCCA–HCC.

In conclusion, the role of necroptosis in CCA is complex and multifaceted. While it can act as a protective mechanism against tumor development, it can also contribute to the inflammatory response and promote tumor progression through angiogenesis, cell proliferation, and metastasis [184,185]. The specific cell death type can also shape the tumor surroundings and influence lineage commitment in liver tumorigenesis. Although limited, some initial data suggest a positive role of necroptosis in iCCA, highlighting the need for further research to better understand the primary molecular mechanisms and potential therapeutic consequences of targeting necroptosis in CCA. Figure 4 shows the main molecular pathways of necroptosis in CCA.

### 5.2. Possible Therapeutical Approaches

The primary treatment option for CCA patients is surgery, which can be performed in approximately 30% of patients, mainly those with iCCA. However, recurrence rates following resection remain high, ranging from 50% to 60%, and five-year survival rates are less than 45% [13,187]. Another therapeutic option is orthotopic liver transplantation after neoadjuvant chemotherapy, which is gradually becoming the standard of care for localized iCCA, but is only available at specialized centers for highly selected patients. Moreover, this procedure is contraindicated in the presence of comorbidities or the neoplastic involvement of regional lymph nodes, and the disease-free survival rates range from 23% to 65%, depending on tumor localization (12–36 months) [13]. Due to the desmoplastic stroma and genetic heterogeneity of this type of tumor, treatment resistance can occur, and new biomarkers or targets must be identified. MLKL, RIPK1, and RIPK3, which make up the necrosome, play important roles in regulating this cell death, with MLKL playing the role of the necroptosis executor.

Stimulating or manipulating necroptosis as part of anticancer therapies is a potential and promising approach for overcoming natural or acquired resistance to cell death, and it presents an alternative method for eradicating cancer cells that have developed resistance to other forms of treatment. There are several methods that could be used to induce necroptosis as a therapeutic agent.

**Natural compounds**, like shikonin, have been proven to be effective in inducing necroptosis in tumor cells. According to Han W et al., shikonin, a naturally occurring naphthoquinone, can induce necroptosis, preventing resistance to cancer drugs mediated by drug transporters or antiapoptotic Bcl-2 proteins in human leukemia cell lines [188]. Other natural compounds, such as staurosporine (STS) and neoalbaconol (NA), induce necroptosis in cancer cells when apoptosis is inhibited [189]. STS, for example, induces RIPK1 and MLKL-dependent necroptotic cell death in leukemia cells when caspases are inactivated, while NA promotes necroptosis in cancer cells by remodeling cellular energy metabolism [190,191].

**The kinase inhibitor** Bl2536 is a compound that inhibits the kinase that regulates mitotic progression, polo-like kinase 1 (Plk1). Bl2536 has been shown to induce mitotic catastrophe in prostate cancer cells that do not respond to androgen, by inhibiting Plk1, ultimately leading to necroptotic cell death. Another compound, Compound C (dorsomorphin), selectively inhibits AMP-activated protein kinase, preventing the phosphorylation of RIPK1 and RIPK3 in glioma cells, thus promoting necrosome formation [192,193,194,195]. Fas ligand, also known as CD95 ligand (CD95L), is a molecule that induces apoptosis. However, in a study by Geserick P et al., it was found to also induce necroptosis by downregulating cIAPs, which are responsible for the polyubiquitination of RIPK1 and direct the pathway towards NF-κB signaling [196,197].

**Oncolytic viruses** are another way to induce necroptosis as part of immunogenic cancer cell death (ICD). During ICD, cancer cells undergo necroptosis and expose calreticulin on their surface, as well as release ATP, HMGB1, DAMPs, and PAMPs, which activate antitumor immunity. These viruses can be genetically modified to specifically target and infect cancer cells, inducing them to undergo ICD and die, while sparing healthy cells [198].

**Metal nanoparticles**, such as silver nanoparticles (AgNPs), have been found to stimulate various types of cell death, such as necroptosis, apoptosis, and autophagy, in pancreatic cancer cells. The consequence is a decreased cell proliferation and viability because the levels of p53, a tumor suppressor protein, and also proteins involved in necroptosis and autophagy, are higher [199]. Selenium nanoparticles (SeNPs) also induce necroptosis through the generation of ROS in a prostate adenocarcinoma cell line. This cell line is dependent on RIP1, but independent of necrosome formation. Smac mimetics, which inhibit the cellular inhibitor of apoptosis proteins (cIAPs), bypass apoptosis resistance by inducing necroptosis, and increase TNFα-induced cell death in apoptosis-resistant cells. Smac mimetics require ROS for their activity and enhance RIPK1/RIPK3 necrosome stabilization. They can sensitize cholangiocarcinoma cells to standard chemotherapy with gemcitabine. Proteasome inhibitors, such as MG132 and Bortezomib, cause necroptosis by RIPK3/MLKL binding, and Obatoclax, a small molecule that inhibits Bcl-2 proteins, triggers necroptosis by linking necrosome and autophagosomes. PolyI:C, a viral dsDNA analog, induces necroptosis in colon cancer together with zVAD, equally in vivo and in vitro. Matrine, an alkaloid, induces necroptosis in CCA cell lines too, which is different from its principal role of stimulating apoptosis in many other types of cancer cells [200,201].

Based on the current research, it appears that targeting necroptosis may be a promising method for anti-cancer therapies, especially for CCA. Several agents, such as natural compounds, chemotherapeutic drugs, oncolytic viruses, metal nanoparticles, and small molecules, have been found to induce necroptotic cellular death. Additionally, there are ongoing efforts to develop specific and stable pharmaceutical combinations that can enhance the efficacy of necroptosis-inducing agents. Therefore, targeting necroptosis is a potential therapeutic strategy that warrants further investigation, and has the potential to improve cancer treatment, including for CCA.

## 6. Conclusions

Programmed cell death plays a crucial role in the pathogenesis and treatment of CCA. The dysregulation of apoptotic and non-apoptotic cell death pathways such as ferroptosis, pyroptosis and necroptosis contributes to CCA development and progression. Furthermore, these pathways can be targeted by novel therapeutic strategies, including the use of specific inhibitors and Smac mimetics, as well as oncolytic viruses and metal nanoparticles.

Personalized medicine approaches [202,203,204,205,206,207,208,209,210], such as genomic profiling and the identification of biomarkers, may help to identify specific vulnerabilities in CCA cells and guide the selection of optimal therapeutic agents capable of modulating the cell death programs. In this scenario, mutations in TP53 [143,211,212,213,214] and KRAS genes have been associated with poor prognosis and resistance to chemotherapy, while mutations in IDH1 and FGFR2 genes may predict responses to targeted therapies.

Overall, a better understanding of the complex interplay between different cell death pathways and their regulation in cholangiocarcinoma is needed to develop effective and personalized treatments for this aggressive cancer.

## Figures and Tables

**Figure 1 cancers-15-03638-f001:**
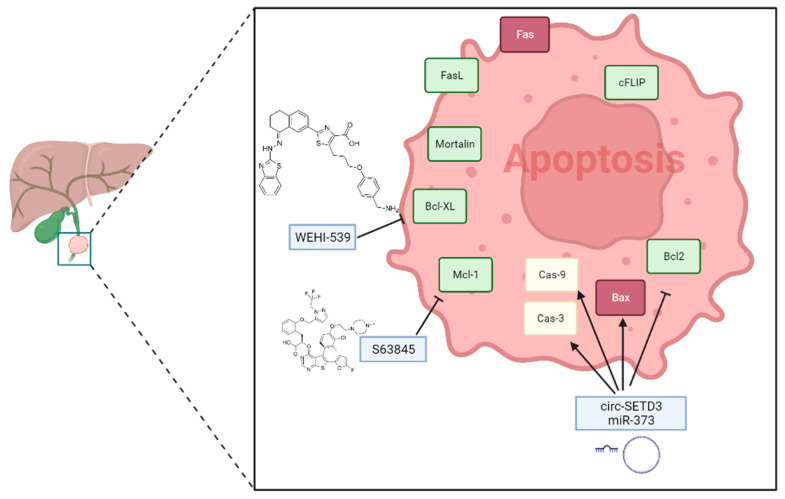
The scheme highlights the main molecular pathways of apoptosis in CCA. The depicted pathways highlight the intricate interplay of various molecules, including pro-apoptotic proteins such as Bax and Bak, anti-apoptotic proteins like Bcl-2 and Bcl-xL, caspases, and FAS. FAS, a cell surface receptor, initiates the extrinsic pathway of apoptosis upon binding with its ligand, FASL. Image created with the support of Biorender.com (url https://www.biorender.com/).

**Figure 2 cancers-15-03638-f002:**
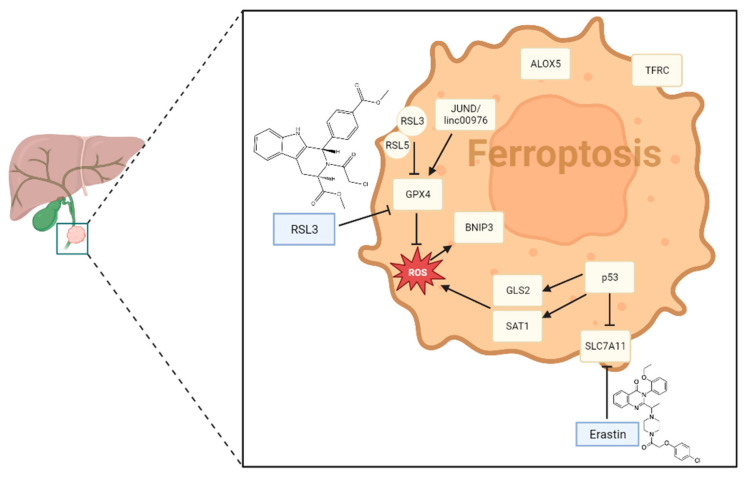
The scheme highlights the main molecular pathways of ferroptosis in CCA. The depicted pathways highlight the crucial molecular players, including lipid peroxidation, glutathione metabolism, iron metabolism, redox homeostasis, and the involvement of Erastin, ALOX5, and BNIPe. Erastin inhibits the cysteine–glutamate transporter system, leading to glutathione depletion and increased susceptibility to ferroptosis. ALOX5, an enzyme involved in the synthesis of lipid mediators, promotes lipid peroxidation and contributes to ferroptosis. BNIPe, a member of the Bcl-2 family, induces mitochondrial dysfunction and enhances susceptibility to ferroptosis. Understanding these molecular mechanisms, including the roles of Erastin, ALOX5, and BNIPe, is crucial for developing targeted therapeutic strategies to induce ferroptosis in CCA, offering a potential avenue for cancer treatment. Image was created with the support of Biorender.com (https://www.biorender.com/).

**Figure 3 cancers-15-03638-f003:**
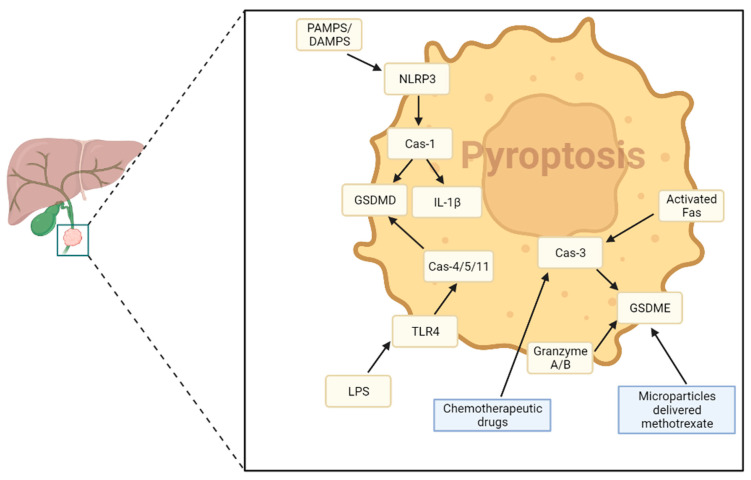
The scheme highlights the main molecular pathways of pyroptosis in CCA. The depicted pathways highlight the intricate interplay of various molecules, including inflammasomes, caspases, gasdermin proteins (GSDMD), and pro-inflammatory cytokines such as IL-1β. Inflammasomes, multiprotein complexes, serve as sensors of cellular stress and activate caspases, particularly caspase-4/5/11, initiating pyroptosis. Gasdermin proteins are responsible for forming pores in the the cell membrane, leading to cell swelling and the release of pro-inflammatory cytokines. Understanding these molecular mechanisms is essential for developing targeted therapies to induce pyroptosis in CCA, potentially offering a novel approach for cancer treatment with implications for inflammation-associated diseases. Image created with the support of Biorender.com (https://www.biorender.com/).

**Figure 4 cancers-15-03638-f004:**
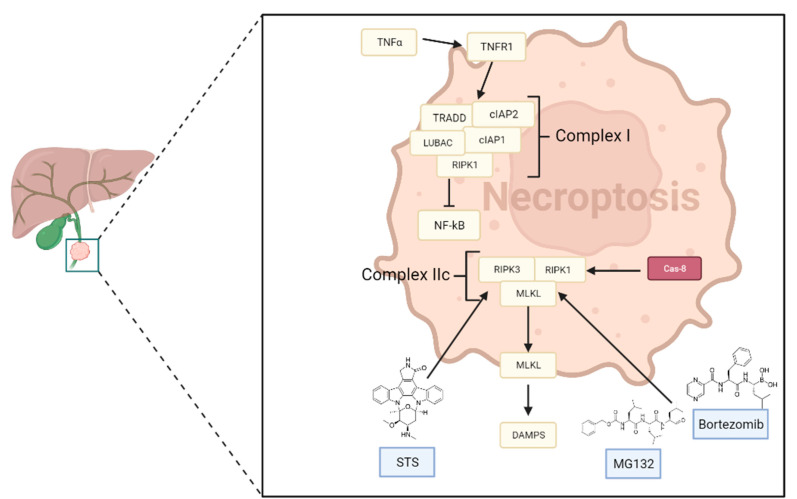
The scheme highlights the main molecular pathways of necroptosis in CCA. The depicted pathways highlight the intricate interplay of various molecules, including receptor-interacting protein kinases (RIPKs), mixed lineage kinase domain-like protein (MLKL), and downstream effector proteins. In necroptosis, the activation of RIPKs, particularly RIPK1 and RIPK3, triggers the formation of a necrosome complex, leading to MLKL phosphorylation and subsequent membrane rupture. The release of intracellular contents stimulates an inflammatory response. Understanding these molecular mechanisms is essential for developing targeted therapies to induce necroptosis in CCA, potentially offering a novel strategy for cancer treatment with implications for modulating cell death pathways. Image created with the support of Biorender.com (url https://www.biorender.com/).

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
