# Peer review of "Programmed Cell Death Pathways in Cholangiocarcinoma: Opportunities for Targeted Therapy"

_cancers, 2023, doi:10.3390/cancers15143638_

Round 1
Reviewer 1 Report
Please, clarify why TOR CENTRE is among the authors, what does it mean?
Subsections in the Introduction, Signaling pathways and Therapeutical approaches would be helpful.
English is understandable, but there is a lot of verbosity i.e. many sentences are too long and unclear.
Tables should contain bullet points i.e. shorter sentences
There is a lot of verbosity.
Author Response
Ref: cancers-2428473
"Programmed Cell Death Pathways In Cholangiocarcinoma: Opportunities For Targeted Therapy"
Submitted to: Cancers
Before we begin the point-by-point review of the list of concerns, we would like to thank the Reviewer for their comments on how to improve the manuscript, which has been revised accordingly, as well as the Editors for calling for a new submission of an improved version of our manuscript.
Reviewer#1
Please, clarify why TOR CENTRE is among the authors, what does it mean?
Reply: The Tor Vergata Oncoscience Research (TOR) Centre is an interdepartmental research center of excellence within the University of Rome Tor Vergata. The inclusion of TOR Centre among the authors suggests that members of the TOR consortium played a role in writing the review. The list of TOR consortium members can be found on the title page.
Subsections in the Introduction, Signaling pathways and Therapeutical approaches would be helpful.
Reply: thanks for this point out. We added a Signaling pathways and Therapeutical approaches subsection in the Introduction.
English is understandable, but there is a lot of verbosity i.e. many sentences are too long and unclear.
Reply: A text revision has been conducted with the aim of reducing verbosity.
Tables should contain bullet points i.e. shorter sentences
Reply: In our manuscript, bullet points are presented in boxes at the end of each paragraph.
Reviewer 2 Report
Dear colleagues,
thank you for the comprehensive summary and review of opportunities for targeted therapy of cholangiocellular cancer.
You give a good overview of programmed cell death pathways and potential targeting cholangiocellular cancer,
however, many aspects , eg. miRNA-approaches, are still a few steps away from clinical relevance.
I would also suggest some minor revisions:
1.
In lines 268-275 the role of overexpression of miR-373 does not become clear. Lines 271-272 state the opposite of lines 272-273.
So this paragraph (lines 268-275) should be reconsidered and rewritten.
2.
In lines 386-388 it does not become entirely clear how BNIP correlates with immune cells (negative or positive correlation) or whether BNIP alters infiltration of the tumor by NK or neutrophil CD56 cells.
In lines 389-391 it does not become clear what risk score has a negative effect on IFN-gamma release.
This paragraph (lines 380-397) should also be reconsidered an rewritten.
Best regards
Author Response
Ref: cancers-2428473
"Programmed Cell Death Pathways In Cholangiocarcinoma: Opportunities For Targeted Therapy"
Submitted to: Cancers
Before we begin the point-by-point review of the list of concerns, we would like to thank the Reviewer for their comments on how to improve the manuscript, which has been revised accordingly, as well as the Editors for calling for a new submission of an improved version of our manuscript.
Reviewer#2
thank you for the comprehensive summary and review of opportunities for targeted therapy of cholangiocellular cancer.
You give a good overview of programmed cell death pathways and potential targeting cholangiocellular cancer,
however, many aspects , eg. miRNA-approaches, are still a few steps away from clinical relevance.
Reply: we would like to thank the Reviewer for expressing interest in our work.
In lines 268-275 the role of overexpression of miR-373 does not become clear. Lines 271-272 state the opposite of lines 272-273.
So this paragraph (lines 268-275) should be reconsidered and rewritten.
Reply: we modified the text accordingly.
In lines 386-388 it does not become entirely clear how BNIP correlates with immune cells (negative or positive correlation) or whether BNIP alters infiltration of the tumor by NK or neutrophil CD56 cells.
In lines 389-391 it does not become clear what risk score has a negative effect on IFN-gamma release.
This paragraph (lines 380-397) should also be reconsidered an rewritten.
Reply: thanks for this point out. In the new version of our manuscript, we reorganized the sentences as indicated, clarifying the role of BNIP3 in NK cell metabolism.
Reviewer 3 Report
Programmed Cell Death Pathways In Cholangiocarcinoma: Opportunities For Targeted Therapy is a comprehensive review of all pathways involved in cell death specific to CCA. It has major sections, apoptosis, ferroptosis, pyroptosis, and necroptosis. In each section, the authors described the main players in the pathway and suggested potential therapeutic agents. They dealt with every molecule in depth and described all the preclinical/clinical evidence available. It will be a good reference article for researchers working on any of the molecules in these pathways. Please consider the following suggestions to make this a better article.
Overall, it is an exhausting article and some sections such as the introduction can be trimmed to 50- 60 lines. Consider moving tables up for each section. Other suggestions and lines numbers are as follows,
- 69 - expand HCC
- 81 – maintain intrahepatic CCA as iCCA
- 80-90 – consider revising classification, starting from iCCA and eCCA, then into peripheral/central vs perihilar and distal.
- I would move Table 1 up to page 4. Add a sentence to the effect of, ‘we summarized the molecules involved in the main signaling pathway in Table 1…’ This would be easy on the readers to follow as there is a lot of text in Section II.I which needs to be read carefully. This applies to all the sections. - References for the following lines are in the middle of the sentence, move it to the end sentence
o 122, 236, 250,355, 391,403, 433, 447, 455, 459, 484, 530, 538, 548, 610, 718, 721
- References are needed for most of the lines in the following lines
o 107 to 120
o 171-189
o 238-241
o 350-354
- Table for microRNA text would help for section II.2
- Some subsections have a Roman numerical (II.I) and some have regular numerical (II.2) – maintain uniformity
- Combine the following lines in the preceding paragraph as they all belong to one study?
o 424-432
o 621-629
- 580-602 – consider moving the references to the first line of the paragraph if appropriate or adding references to relevant lines.
- 616 - HCC was already used in line 69
Author Response
Ref: cancers-2428473
"Programmed Cell Death Pathways In Cholangiocarcinoma: Opportunities For Targeted Therapy"
Submitted to: Cancers
Before we begin the point-by-point review of the list of concerns, we would like to thank the Reviewer for their comments on how to improve the manuscript, which has been revised accordingly, as well as the Editors for calling for a new submission of an improved version of our manuscript.
Reviewer#3
Programmed Cell Death Pathways In Cholangiocarcinoma: Opportunities For Targeted Therapy is a comprehensive review of all pathways involved in cell death specific to CCA. It has major sections, apoptosis, ferroptosis, pyroptosis, and necroptosis. In each section, the authors described the main players in the pathway and suggested potential therapeutic agents. They dealt with every molecule in depth and described all the preclinical/clinical evidence available. It will be a good reference article for researchers working on any of the molecules in these pathways. Please consider the following suggestions to make this a better article.
Reply: we would like to thank the Reviewer for expressing interest in our work, and for their availability to review our manuscript.
Overall, it is an exhausting article and some sections such as the introduction can be trimmed to 50- 60 lines. Consider moving tables up for each section. Other suggestions and lines numbers are as follows,
- 69 - expand HCC
- 81 – maintain intrahepatic CCA as iCCA
- 80-90 – consider revising classification, starting from iCCA and eCCA, then into peripheral/central vs perihilar and distal.
Reply: thanks for this point out. As possible, we trimmed both the introduction and other paragraphs. Also we modified the text as suggested by the reviewer.
- I would move Table 1 up to page 4. Add a sentence to the effect of, ‘we summarized the molecules involved in the main signaling pathway in Table 1…’ This would be easy on the readers to follow as there is a lot of text in Section II.I which needs to be read carefully. This applies to all the sections.
Reply: done
References for the following lines are in the middle of the sentence, move it to the end sentence
o 122, 236, 250,355, 391,403, 433, 447, 455, 459, 484, 530, 538, 548, 610, 718, 721
Reply: where possible we moved the references at the end of the sentences.
- References are needed for most of the lines in the following lines
o 107 to 120
o 171-189
o 238-241
o 350-354
Reply: done
- Table for microRNA text would help for section II.2
Reply: in the new version of the manuscript, we added a table with the main microRNA used as potential Therapeutical Approaches in CCA.
Table 2 promising microRNA as therapeutic target for CCA
|
Molecule |
Function |
References |
|
circSETD3 |
Circular RNA containing multiple miRNA binding sites has been implicated in CCA progression. |
69 |
|
miR-421 |
Promotes cell proliferation in human gastric cancer and represents a promising therapeutic target for CCA treatment. |
70,71 |
|
miR-373 |
its overexpression promotes apoptosis in CCA cells by targeting ULK1 |
72 |
|
miR-191 |
involved in the initiation and progression of CCA |
73 |
- Some subsections have a Roman numerical (II.I) and some have regular numerical (II.2) – maintain uniformity
Reply: thanks for this point out. We corrected the manuscript according to the reviewer suggestion.
- Combine the following lines in the preceding paragraph as they all belong to one study?
o 424-432
o 621-629
Reply: done
- 580-602 – consider moving the references to the first line of the paragraph if appropriate or adding references to relevant lines.
Reply: thanks for this point out. In order to trim down the paper, we deleted this sentence.
- 616 - HCC was already used in line 69
Reply: thanks for this point out.
Reviewer 4 Report
The authors conducted a review article on programmed cell death pathways in cholangiocarcinoma and identifying opportunities for targeted therapy. I have the following comments.
1. Overall, the manuscript offers a comprehensive review of programmed cell death pathways. However, the manuscript would be greatly enhanced if the authors can include schematic figures of the described pathways to visually showing the complex molecular interactions and mechanism of molecular actions. This would be helpful and appealing to readers.
2. I find the extensive written description of molecular pathways somewhat boring and tedious to follow. By including the schematic figures, the article can be more concise with the descriptions and decrease in word length.
3. The current mainstay first line of systemic therapy for cholangiocarcinoma is chemotherapy. This deserves a paragraph of description on the method of cellular death occur for chemotherapy induced cytotoxicity before highlighting more experimental targeted therapies.
4. The immune-mediated cellular response is an area of current research interest with immunotherapy. Although mentioned in the manuscript, the authors should have more emphasis on programmed cell death pathways and possible subsequent immune-direct responses.
5. Genomic architecture of FGFR2 fusions is an emerging area of targeted therapy for cholangiocarcinoma. This deserves to be discussed in the review manuscript in order to be relevant and current with the literature.
Several minor misspellings were found (page 5, line 208; page 13, line 501, etc...)
Author Response
Ref: cancers-2428473
"Programmed Cell Death Pathways In Cholangiocarcinoma: Opportunities For Targeted Therapy"
Submitted to: Cancers
Before we begin the point-by-point review of the list of concerns, we would like to thank the Reviewer for their comments on how to improve the manuscript, which has been revised accordingly, as well as the Editors for calling for a new submission of an improved version of our manuscript.
Reviewer#4
The authors conducted a review article on programmed cell death pathways in cholangiocarcinoma and identifying opportunities for targeted therapy. I have the following comments.
Reply: we would like to thank the Reviewer for expressing interest in our work, and for their availability to review our manuscript.
- Overall, the manuscript offers a comprehensive review of programmed cell death pathways. However, the manuscript would be greatly enhanced if the authors can include schematic figures of the described pathways to visually showing the complex molecular interactions and mechanism of molecular actions. This would be helpful and appealing to readers.
Reply: thanks for this point out. In the new version of the manuscript, In the revised version of the manuscript, we have included two figures illustrating the molecular mechanisms of key mediators involved in apoptosis and ferroptosis, respectively.
- I find the extensive written description of molecular pathways somewhat boring and tedious to follow. By including the schematic figures, the article can be more concise with the descriptions and decrease in word length.
Reply: according to the reviewer suggestion, we incorporated schematic figures in the revised version to improve the overall readability of the article also removing several sentences summarized by the figures.
- The current mainstay first line of systemic therapy for cholangiocarcinoma is chemotherapy. This deserves a paragraph of description on the method of cellular death occur for chemotherapy induced cytotoxicity before highlighting more experimental targeted therapies.
Reply: Thank you for your comment. We appreciate your suggestion to include a paragraph describing the method of cellular death occurring with chemotherapy-induced cytotoxicity before discussing experimental targeted therapies. We agree that providing an overview of the mechanism of cellular death induced by chemotherapy is important for contextualizing the subsequent discussion on targeted therapies. In the revised version of the manuscript, we have incorporated a dedicated paragraph that outlines the cellular death mechanisms associated with chemotherapy, thereby providing a comprehensive understanding of the current mainstay therapy for cholangiocarcinoma.
Pag. 3
Chemotherapy-induced cytotoxicity in CCA involves a complex interplay of molecu-lar mechanisms that ultimately lead to cellular death [23-30]. One of the primary mecha-nisms is through the activation of apoptotic pathways. Chemotherapeutic agents, such as cisplatin and gemcitabine, can trigger a cascade of events that culminate in the activation of caspases, which are key executioners of apoptosis [29]. These caspases, particularly caspase-3, play a crucial role in cleaving various cellular substrates, including structural and functional proteins, leading to cellular dismantling and programmed cell death [29]. Additionally, chemotherapy can induce DNA damage, resulting in the activation of DNA repair pathways or, if the damage is irreparable, triggering apoptosis through p53-mediated signaling [24,25]. Another mechanism involved in chemotherapy-induced cytotoxicity is the generation of reactive oxygen species (ROS) within cancer cells [24,25]. Increased ROS levels disrupt cellular redox balance and can cause oxidative stress, lead-ing to DNA damage, lipid peroxidation, and mitochondrial dysfunction, ultimately pro-moting cell death. Understanding these cellular death mechanisms induced by chemo-therapy in CCA is essential for optimizing treatment strategies and exploring novel ther-apeutic approaches.
- The immune-mediated cellular response is an area of current research interest with immunotherapy. Although mentioned in the manuscript, the authors should have more emphasis on programmed cell death pathways and possible subsequent immune-direct responses.
Reply: We agree with the reviewer that the immune-mediated cellular response is a crucial aspect of current research interest, particularly in the context of immunotherapy. According to the reviewer suggestion, we have made the following modifications to the text:
Pag. 7
Additionally, apoptosis can be activated by a third pathway that involves immune cells directly. In fact, the immune-mediated cellular response plays a pivotal role in maintaining homeostasis, eliminating pathogens, and preventing the development of ma-lignancies [48]. In recent years, the field of immunotherapy has gained significant atten-tion as a promising approach for treating various diseases, including cancer. One crucial aspect of the immune-mediated cellular response is the apoptosis. In the context of im-munotherapy, programmed cell death pathways have garnered significant interest due to their potential as targets for therapeutic intervention [48]. The discovery of immune checkpoint inhibitors, such as antibodies against programmed cell death protein 1 (PD-1) and its ligand PD-L1, has revolutionized cancer treatment by unleashing the immune system's ability to recognize and destroy tumor cells [18, 48,49]. By blocking the inhibitory signals mediated by PD-1/PD-L1 interactions, these therapies restore the effector functions of cytotoxic T cells and enhance the immune-mediated cellular response against cancer cells.
- Genomic architecture of FGFR2 fusions is an emerging area of targeted therapy for cholangiocarcinoma. This deserves to be discussed in the review manuscript in order to be relevant and current with the literature.
Reply: We appreciate your suggestion to include a dedicated section discussing the genomic architecture of FGFR2 fusions and their implications for targeted therapy in cholangiocarcinoma. While we recognize the importance of this topic, we regret to inform you that we are unable to incorporate it into the revised version of our manuscript.
The primary focus of our review manuscript is to provide a comprehensive overview of the current understanding of the Programmed Cell Death Pathways In cholangiocarcinoma and its therapeutic approaches. While the genomic architecture of FGFR2 fusions is undoubtedly a relevant and emerging area of research, we believe that an in-depth analysis of this specific topic would exceed the scope and objectives of our review.
Reviewer 5 Report
The manuscript is well organized and written. The data is clearly presented for the reader. Adequate introduction for particular sections allows an accessible presentation of molecular aspects of each pathways. I found it as a valuable review also for readers without extensive basic science background.
I have two minor comments:
1. Verse 558- should be "necroptosis" instead of "ferroptosis"
2. Verse 648- please precise of how long disease-free survival we are talking about
Author Response
Ref: cancers-2428473
"Programmed Cell Death Pathways In Cholangiocarcinoma: Opportunities For Targeted Therapy"
Submitted to: Cancers
Before we begin the point-by-point review of the list of concerns, we would like to thank the Reviewer for their comments on how to improve the manuscript, which has been revised accordingly, as well as the Editors for calling for a new submission of an improved version of our manuscript.
Reviewer#5
The manuscript is well organized and written. The data is clearly presented for the reader. Adequate introduction for particular sections allows an accessible presentation of molecular aspects of each pathways. I found it as a valuable review also for readers without extensive basic science background.
Reply: we would like to thank the Reviewer for expressing interest in our work, and for their availability to review our manuscript.
I have two minor comments:
- Verse 558- should be "necroptosis" instead of "ferroptosis"
- Verse 648- please precise of how long disease-free survival we are talking about
Reply: done
Round 2
Reviewer 3 Report
Authors made necessary changes to the manuscript. Added tables and references.
Author Response
Ref: cancers-2428473
"Programmed Cell Death Pathways In Cholangiocarcinoma: Opportunities For Targeted Therapy"
Submitted to: Cancers
Before we begin the point-by-point review of the list of concerns, we would like to thank the Reviewer for their comments on how to improve the manuscript, which has been revised accordingly, as well as the Editors for calling for a new submission of an improved version of our manuscript.
Reviewer#3
Authors made necessary changes to the manuscript. Added tables and references.
Reply: Thank you for your feedback and suggestions on our manuscript. We appreciate your careful review.
Reviewer 4 Report
Thank you for making the revisions according to my prior comments. I have additional comments.
1. The authors included 2 figures showing the schematic of molecular pathways of apoptosis and ferroptosis. Yet, the figure legends are poorly described and very hard to follow.
2. There have been extensive changes, reorganization made to the revised manuscript that it is essentially a new manuscript. All the editing has made the revision very hard to read. Please include a version of the revised manuscript without the edits that has been done.
3. The manuscript is still very lengthy and need to be more concise, with the use of more schematic figures.
4. The references did not reflect the changes made in the manuscript.
Moderate editing of English language required.
Author Response
Ref: cancers-2428473
"Programmed Cell Death Pathways In Cholangiocarcinoma: Opportunities For Targeted Therapy"
Submitted to: Cancers
Before we begin the point-by-point review of the list of concerns, we would like to thank the Reviewer for their comments on how to improve the manuscript, which has been revised accordingly, as well as the Editors for calling for a new submission of an improved version of our manuscript.
Reviewer#4
- The authors included 2 figures showing the schematic of molecular pathways of apoptosis and ferroptosis. Yet, the figure legends are poorly described and very hard to follow.
Reply: Dear Reviewer,
We appreciate your feedback and thank you for bringing up the issue regarding the figure legends in our manuscript. We have carefully considered your comment and have made significant improvements to the figure legends to enhance their clarity and comprehension.
In particular, we have extended the figure legends of both figures, providing more detailed explanations and ensuring that they effectively guide the readers through the schematic of the molecular pathways of apoptosis and ferroptosis. We have incorporated concise yet informative descriptions that highlight the key components and interactions within each pathway, making it easier for readers to understand the underlying mechanisms.
Moreover, in the new version of our manuscript, we added 2 figures concerning the main molecular pathways of pyroptosys and necroptosis in CCA.
- There have been extensive changes, reorganization made to the revised manuscript that it is essentially a new manuscript. All the editing has made the revision very hard to read. Please include a version of the revised manuscript without the edits that has been done.
Reply: To address your concern and facilitate a better understanding of the revisions made, we have uploaded a clean version (supplementary) of the revised manuscript without the edits.
- The manuscript is still very lengthy and need to be more concise, with the use of more schematic figures
Reply: We appreciate your feedback on the length of the manuscript. While we understand your concern regarding its length, we would like to inform you that the paragraphs have already been revised based on the suggestions provided by the other four reviewers. Consequently, it is not possible to significantly reduce the length of the paragraphs without compromising the clarity and scientific integrity of the content.
However, we have taken your suggestion into consideration and have addressed it by adding two additional schematic figures to the manuscript.
- The references did not reflect the changes made in the manuscript.
Reply: Thank you for your comment regarding the references in our manuscript. We apologize for any inconsistencies or discrepancies in the references and their alignment with the changes made in the manuscript.
We have carefully reviewed the references and have made the necessary updates to ensure they accurately reflect the content and changes made in the revised manuscript.
List of added references
1. PMID: 37084797
2. PMID: 36661679
3. PMID: 37324940
4. PMID: 37293061
5. PMID: 37292215
6. PMID: 37289037
7. PMID: 34203270
8. PMID: 25936818
9. PMID: 33428453
10. PMID: 28984478
11. PMID: 27879653
12. PMID: 30827759
13. PMID: 32485280
14. PMID: 33850005
15. PMID: 35912211
Round 3
Reviewer 4 Report
The authors made appropriate revisions to the original manuscript. I find the manuscript informative on the subject content. The figures enhance the description.
There are still a few misspellings in the manuscript, advise the authors to proofread again.
Minor editing of English language required, a few misspelled words.
Author Response
The authors made appropriate revisions to the original manuscript. I find the manuscript informative on the subject content. The figures enhance the description.
There are still a few misspellings in the manuscript, advise the authors to proofread again.
Minor editing of English language required, a few misspelled words.
Reply: Thank you for your suggestions. We have carefully considered your comments and made the necessary revisions to improve the clarity and accuracy of the manuscript.